# Identification of Factors Affecting Environmental Contamination Represented by Post-Hatching Eggshells of a Common Colonial Waterbird with Usage of Artificial Neural Networks

**DOI:** 10.3390/s22103723

**Published:** 2022-05-13

**Authors:** Agnieszka Sujak, Dariusz Jakubas, Ignacy Kitowski, Piotr Boniecki

**Affiliations:** 1Department of Biosystem Engineering, Faculty of Environmental Engineering and Mechanical Engineering, University of Life Sciences in Poznań, Wojska Polskiego 50, 60-627 Poznań, Poland; piotr.boniecki@up.poznan.pl; 2Department of Vertebrate Ecology and Zoology, Faculty of Biology, University of Gdańsk, Wita Stwosza 59, 80-308 Gdańsk, Poland; biodj@univ.gda.pl; 3Department of Zoology and Animal Ecology, University of Life Sciences in Lublin, Akademicka 13, 20-950 Lublin, Poland; ignacy.kitowski@up.lublin.pl

**Keywords:** biomaterial, biomonitoring, grey heron, elemental analysis, artificial neural networks

## Abstract

Artificial Neural Networks are used to find the influence of habitat types on the quality of the environment expressed by the concentrations of toxic and harmful elements in avian tissue. The main habitat types were described according to the Corine Land Cover CLC2012 model. Eggs of free-living species of a colonial waterbird, the grey heron *Ardea cinerea,* were used as a biological data storing media for biomonitoring. For modeling purposes, pollution indices expressing the sum of the concentration of harmful and toxic elements (multi-contamination rank index) and indices for single elements were created. In the case of all the examined indices apart from Cd, the generated topologies were a multi-layer perceptron (MLP) with 1 hidden layer. Interestingly, in the case of Cd, the generated optimal topology was a network with a radial basis function (RBF). The data analysis showed that the increase in environmental pollution was mainly influenced by human industrial activity. The increase in Hg, Cd, and Pb content correlated mainly with the increase in the areas characterized by human activity (industrial, commercial, and transport units) in the vicinity of a grey heron breeding colony. The decrease in the above elements was conditioned by relative areas of farmland and inland waters. Pollution with Fe, Mn, Zn, and As was associated mainly with areas affected by industrial activities. As the location variable did not affect the quality of the obtained networks, it was removed from the models making them more universal.

## 1. Introduction

Biomaterials, such as various avian tissues, are widely used for sensing and monitoring the levels of environmental pollution as a biological data storing media [1,2]. Eggshells of different species of waterbirds, including herons *Ardeidae,* are often used to determine the health of aquatic ecosystems [3,4,5,6,7]. The grey heron *Ardea cinerea* is a highly opportunistic predator, with a diet varying widely according to habitat and season; depending on location, it may be dominated by fish, crustaceans, or mammals [8,9]. They can also forage in non-aquatic habitats, such as pastures, to supplement their basic diet (fish) with invertebrates and mammals [10]. Being a top predator in freshwater ecosystems, the grey heron is considered an important ‘indicator species’ in environmental monitoring, as it can biomagnify and bioaccumulate toxic and non-toxic elements, pesticides, and various pollutants [11,12]. Since the grey heron adopts the income breeder strategy for acquiring nutrients for egg production [13,14], their maternal investments (including eggshells) correspond to the contamination of the herons’ foraging areas on breeding grounds [15]. The existence of the unique mechanism of deposition of the harmful elements in the eggshells makes them good sensors of the quality of the environment at nesting sites as the composition of an eggshell represents a relatively short exposure time from a very strictly defined area exploited by the birds during the initiation of the breeding season [7,16,17].

In search of food, grey herons usually travel between 1–10 km from the nesting site, preferring smaller distances up to 10 km. When attractive food is found at a greater distance, they decide to fly longer distances, up to 20 km [8,18,19], which can be assumed to be a sensing range.

In this study the environmental quality was studied with the use of artificial neural networks (ANNs).

In recent years, neural models have started to play a crucial role in the natural sciences. They are able to solve a number of issues, including the so-called scientific problems defined as unstructured, i.e., problems that cannot be algorithmized and for which there is insufficient knowledge or experimental data. Neural modeling is a process of creation, verification, and exploitation of generated artificial neural networks (ANNs). The main factor distinguishing neural models, when compared to symbolic methods of machine learning (e.g., rules or decision trees), is a totally different way of representing the knowledge obtained by the system during the learning process. This representation, called post-symbolic, is a direct consequence of the computational implementation used and the neural model topology adopted [20]. Among the most important and desirable properties of neural models are: high computational power, reliability and resistance to noise, simplicity of use, ease of construction, ability to generalize the acquired knowledge, and biological inspiration.

The generated ANN topologies are, among others, a universal and convenient instrument increasingly used in scientific research. They are often used as predictive instruments [21,22]. In contrast to other analytical methods, such as linear models, ANNs are not sensitive to multi-collinearity or non-normal distribution (collinearity). Other algorithms of binary regression instruments, dedicated to solving various prediction problems, such as decision trees, SVM (Support Vector Machine) models, rules (IF…THEN…), etc., are frequently used.

ANNs are more and more frequently applied to the analysis of the results of experimental research conducted in many research programs [23,24,25,26,27]. Neural modeling can be effectively used to solve classification and regression problems in biological sciences, including prediction [28,29]. The increasing popularity of ANNs in environmental monitoring and analyses is due to the fact that this tool can be utilized to model non-linear and complex phenomena even without full knowledge about the underlying changing mechanisms [30].

The choice of neural prediction techniques was motivated, among others, by their high performance (quality of generated forecasts), universality, low cost, and ease of use. The advantage was also fewer assumptions than in regression/linear models).

The aim of the study was to check the effectiveness of ANNs in the assessment of the levels of eggshell contamination on the basis of the habitat composition (types of land cover) around the grey heron colonies (in a radius up to 20 km) representing their foraging grounds. Precise and accurate ANNs may serve as useful tools for environmental monitoring enabling the determination of areas with a high risk of heavy metal accumulation in living organisms and/or biomagnification. It is especially important in the case of colonially breeding waterbirds, such as grey herons, transferring minerals, nutrients and pollutants from aquatic to terrestrial ecosystems [31,32].

In this study, an attempt was made to answer the following questions:Are ANNs an adequate tool to support the proposed multi-contamination rank index (CONT_IN; see detail definition in Materials and Methods) or a specific element concentration in an estimation process?Which of the selected input descriptors of the developed ANNs (habitat types around breeding colony) are representative features during the CONT_IN or specific element determination?Which ANN topology is optimal for the CONT_IN or specific element concentration identification process?

## 2. Materials and Methods

### 2.1. Collection of Grey Heron Eggshells

Post-hatched eggshells of the grey heron were collected from 22 heronries throughout Poland (Figure 1) characterized by a different composition of habitats with a 20 km radius around the colony (representing a range of foraging flights [33]). Eggshells were collected randomly from the ground underneath the trees with nests at the end or after the chick-rearing period, i.e., in June–July 2015. The content of 18 elements was measured in 4–12 eggshells from each localization (depending on availability). The following numbers of grey heron post-hatching eggshells were analyzed in particular colonies (colony codes used in Figure 1 in brackets): 12 in Gardzka Kępa (GK), 12 in Kąty Rybackie (KR), 12 in Brwilno (BR), 12 in Chutkowice (CH), 12 in Kiersity (KS), 12 in Gaładuś (GA), 12 in Gołdap (GO), 10 in Osiek (OS), 12 in Sławskie lake (JS), 11 in Mosty (MO), 12 in Zborowskie (ZB), 12 in Dzierżno Duże Reservoir (DD), 12 in Jawory (JA), 10 in Pogorzałki (PO), 12 in Malczyce (MA), 12 in Limajno Lake (LI), 12 in Kiełcz (KI), 12 in Raszyn (RA), 12 in Otmuchów (OT), 12 in Stobrawa (ST), 12 in Płoskinia (PL), and 4 in Wrocław ZOO (WR).

### 2.2. Concentration of Elements in Post-Hatching Eggshells

Inductively Coupled Plasma Optical Emission Spectrometry (ICP-OES) was used to estimate concentrations of elements in the eggshells according to the procedure described previously [7,34]. The elemental concentration (Al, As, Cd, Cr, Cu, Fe, Hg, Mn, Mo, Ni, Pb, Sc, Se, Sr, V, and Zn) was analyzed in 4–12 eggshells per colony. Data on elemental concentration for 7 colonies were also presented [7]. 

### 2.3. Multi-Contamination Rank Index (CONT_IN)

To illustrate the eggshell contamination level for a particular colony in one index, we proposed the multi-contamination rank index (CONT_IN). Firstly, the mean concentrations of all the examined elements apart from Ca and Mg were calculated. Then, the studied colonies were ranked according to the decreasing mean concentration of the analyzed elements. The colony with the highest mean concentration value for an examined element was assigned the number 1 and the lowest 22 (as many numbers were used as the number of locations). The operation was repeated for all elements. CONT_IN was defined as the sum of all ranks assigned to all the considered elements for a certain heronry.

No data exist on the level of toxicity of the elements based on their concentration in eggshells. To illustrate the levels of toxic elements in eggshells from a particular colony in one index, we proposed the above multi-contamination rank index. The value of the created coefficient is conceptional (original authors’ idea); the multi-contamination rank index with the highest value represents the lowest toxicity level, while its lowest value represents the highest toxicity level.

### 2.4. Type of Land Use—Habitat Analyses

To identify the main habitat types (type of land use) in the 20 km buffers around the studied heronries (reflecting the longest range in foraging flights [33]), main habitat types were determined based on the level 2 data from the CORINE Land Cover (CLC) 2012, version 18_5. CLC uses a minimum mapping unit of 25 ha, and land cover classes are grouped in a three-level hierarchy with an ascending number of land cover classes from 5 to 44 (https://land.copernicus.eu/pan-european/corine-land-cover/clc-2012 (accessed on 1 February 2022)). Due to the relatively small sample size, we used level 2 with 14 types of land covers distinguished (listed in Table 1). Spatial analyses (the extraction of particular landscape features) were performed with the application of the ArcMap software, version 10.3.1 (ArcGIS, ESRI, Redlands, CA, USA). In the performed analyses, we used relative area of particular habitat types in 20 km radius buffers around colonies summing to 1. In the case of the Gołdap colony (GO) with a buffer area including partly Russia not covered by CORINE model only the area in Poland was considered. The above analysis allowed for the collection of an empirical database describing the type of land use within a 20 km radius from examined heronries (See Appendix A). 

### 2.5. Simulation Studies

To create ANNs, we used a dataset consisting of a series of measurements of 14 representative features (constituting the types of land use analyzed within a buffer with a 20 km radius around the studied heronries), which could have an impact on the levels of contaminants identified in the eggshells (Table 1) expressed as mean values of element concentrations. The structure of the input set was adjusted to the requirements of the ANN simulator implemented in the commercial STATISTICA package (v13.3, Statsoft, Cracow, Poland). Each learning case consisted of 14 input variables (descriptors) and one output variable. Each output variable was analyzed in a separately created network (input variables (1–14) and output variables (A–H)—according to Table 1).

Although the ANN simulator was focused on 4 standard ANN topologies: Linear, GRNN (Generalized Regression Neural Network), MLP (Multi-layer Perceptron), and RBF (Radial Basis Function), the generated topologies of the multi-layer perceptron type (MLP with 1 hidden layer) were hybrid taught by the BP (Back Propagation) algorithm and then by the CG (Conjugate Gradient) algorithm. The generated network with radial basis functions (RBF type) was taught in 3 stages: the KM (K-Means: K-means), KNN (K-Nearest Neighbor: K-nearest neighbors), and PI (pseudoinversion and linear least squares optimization) methods. In order to identify the significance level of individual input variables on the functioning of the generated ANN, a network sensitivity analysis was performed using the standard procedure implemented in the STATISTICA package. To interpret the results of ANN, i.e., to know the direction of the relationship between response variables and input variables, the Pearson correlation coefficient was calculated.

The structure of the learning set was represented by an empirical data set containing 44 measurement cases. Each learning case included 14 input variables (descriptors) and 1 output variable, relevant to the predicted variable. The collection was subdivided into 3 subsets: training containing 33 cases, validation containing 5 cases, and test containing 6 cases. The non-symmetrical division of training cases (33:5:6) was justified by the limited amount of empirical data (in relation to the number of descriptors), which resulted from the data accessibility and the high cost of conducted research. The training, validation and test cases that form part of the learning set were subjected to standard pre-processing procedures (implemented in the STATISTICA package in the ST Neural Networks module), involving scaling their values to normalized numerical intervals. The applied pre- and post-processing techniques and the parameters controlling their operation were stored in the ST Neural Networks sub-program in a file containing the definition of SSN. The pre- and post-processing tool available in the SSN Editor was used to create neural models. The minimax function (without data normalization) was used as the conversion function.

ANN was tested and compared with 14 input variables and CONT_IN or a concentration of selected element output variables as an output. A series of neural estimators was automatically created for forecasting the connection of environmental pollution with the type of land use based on the previously defined contamination index or on the mean concentrations of selected toxic and essential elements (separately for each element).

## 3. Results

### 3.1. Concentrations of Elements in the Eggshells and Multi-Contamination Rank Index

The concentration of As in post-hatching eggshells of grey herons from the studied colonies ranged between 0 (single eggshells from the GK, OS, MO, LI, and RA locations) and 1.40 mg/kg with mean values between 0.28 (GK) and 1.07 (MA). Cd ranged between 0 (single eggshells from OS, JS, and ZB) and 57.7 μg/kg (MA) with mean values between 6.4 (GA) and 43.7 μg/kg (PO). Fe ranged between 1.76 (JS) and 85.26 mg/kg (single eggshell from the MO colony), while its mean concentrations amounted to between 3.25 (JS) and 18.89 mg/kg (MO). Hg at detectable levels was found in eggshells from all colonies apart from single samples from KA, GA, and OS heronries. It amounted to between 0 and 0.74 mg/kg (LI) with mean concentrations between 0.08 (GA) and 0.30 mg/kg (LI). Between 0.33 and 27.28 mg/kg of Mn was found respectively for OS and ZB colonies with mean values between 0.59 (PO) and 6.04 mg/kg (OT). Pb ranged between 0.085 and 1.106 mg/kg (both extreme values recorded in the JA colony), while its mean concentration was between 0.32 (GA) and 0.66 mg/kg (KA). In the case of Zn, concentration ranges between 0.09 (GA) and 45.04 mg/kg (single eggshell in JA) were found, while mean values amounted to between 3.39 (GA) and 26.12 (JA). Mean concentrations and ranges for other examined elements including heavy metals in the eggshells of grey herons from the particular studied colonies (in mg/kg of dry mass or in μg/kg when applicable) are presented in Appendix A. 

The metal accumulation pattern/gradient was: Ca > Mg > Sr > Fe > Al > Zn > Mn > Cu > Se >As > Pb > Cr > Hg > V > Ni > Mo > Sc > Cd. For heavy metals, the accumulation pattern/gradient was as follows: Zn > Mn > Cu > Pb > Ni > Cd.

The multi-contamination rank index ranged from 102 to 280 with the highest value (representing the lowest toxic level) being in the JS colony and the lowest value (representing the highest toxic level) in the JA colony (Table 2).

### 3.2. Results Obtained with Use of Computer ANN Simulations

After performing a series of automated computer simulations and creating a set of 60 networks for each considered parameter (executed automatically within the ANN simulator), the optimal neural structure was selected, which proved to be the most effective as a predictive instrument dedicated to the estimation of the CONT_IN indicator, or appropriate indicators expressing the concentrations of toxic elements Hg, Pb, and Cd as well as essential elements as As, Fe, Mn, and Zn.

In the case of the CONT_IN (multi-contamination rank index), the MLP network with the structure 14-16-1 (Figure 2) proved to be optimal. The input layer consisted of 14 neurons with a linear (saturated) activation function and a linear PSP (Post Synaptic Potential) function. The hidden layer consisted of 16 neurons with a logistic activation function and a linear PSP function. The output layer consisted of 1 neuron with a linear activation function (with saturation) and a linear PSP function.

In the case of all the examined indices expressing mean concentrations of elements apart from Cd, the generated topologies were a multi-layer perceptron (with one hidden layer) (Table 3). Interestingly, in the case of Cd, the generated optimal topology was a network with radial basis function (RBF) (Table 4).

The input layer for Cd consisted of 14 neurons with a linear (saturated) activation function and a linear PSP (Post Synaptic Potential) function. Here, the RBF neural network had one hidden layer made of eight radial neurons and one output layer holding a linear neuron. Such a network is characterized by a short learning time and good quality of operation but at the cost of a high complexity of the structure. Networks with radial base functions learn relatively quickly and have the advantage that they never extrapolate functions too far from known data. So, they are in a way the safest. However, as a rule, they are much larger than MLP networks that solve the same problems. This fact makes their operation on the ANN simulating computer more time-consuming.

The generated models were characterized by a low value of the quotient of standard deviations (SD). Its relatively low value for the validation set proves the high quality of the generated neural model. SD for individual subsets ranged from 0.005 to 0.49. This proves that the generated model is resistant to over-fitting (memorizing). This fact is confirmed by the value of the correlation coefficient close to unity (1.00). The values of the indicated parameters of regression statistics reached comparable values for all three subsets (training, validation, and test subsets). This allowed for a conclusion about the generalization ability (acquired in the course of learning) of the generated ANN. The correlation coefficient, defining the correlation between individual input variables for the validation set and the output variable, reached a high value and was between 0.7174 (for Zn) and 0.9998 (for CONT_IN) (Table 3).

The sensitivity analysis of the generated neural models showed a different impact of the type of land use on the level of selected pollutants (Table 4).

The elemental concentration of post-hatching eggshells of grey herons is influenced by the habitat composition in the potential foraging areas around the studied heronries. The analysis showed that the increase in environmental pollution (expressed by a decrease in the CONT_IN value) is mainly influenced by urban fabric and industrial, commercial, and transport units, while the remediation role is played by pastures.

The increase in the Hg, Cd, and Pb content is correlated with the increase in the area occupied by industrial, commercial, and transport units; scrub and/or herbaceous vegetation associations, and by artificial, non-agricultural vegetated areas, while their decrease is conditioned by the presence of inland waters and, surprisingly, by heterogeneous agricultural areas and arable land. In the case of Fe, Mn, Zn, and As, the increase in pollution is associated with an increase in urban fabric, industrial, commercial, and transport units; mines, dumps, and construction sites as well as by scrub and/or herbaceous vegetation associations. A remediation role is played by the presence of pastures.

## 4. Discussion

The present study represents an example of ANN application for environmental monitoring. The generated models were characterized by a low SD value for the validation set that proved their high effectiveness. The performed neural analysis combined with a biological interpretation of data showed that the usage of ANNs was adequate to support both the proposed multi-contamination rank index (CONT_IN) as well as concentrations of toxic and harmful elements.

The sensitivity analysis provides insight into the usefulness of individual input variables during the operation of the generated neural model. It shows the “key” variables important for the optimization process of the ANN that are being created, which must not be omitted. It also indicates the variables that, without losing the quality of the network, may not be taken into account in the process of its creation and used. As indicated by the sensitivity analysis, other considered variables played only a minor role.

Neural models preliminary created with consideration of geographic location did not show dependence of concentrations of particular elements on this parameter. The sensitivity analysis of the individual networks showed that the rank of the location descriptor was 15 (the least important of the 15 variables), showing that this variable does not affect the quality of the obtained networks (not shown). Therefore, this variable was removed from the model. We concluded that if this variable is not necessary to the model, then it becomes location-independent, and thus the issue can be considered based on the characteristics of the types of land use.

The topology of an ANN can be thought of in terms of a directed weighted graph. A layered layout is used, i.e., one in which neurons are divided into several subsets called layers, within which neurons have no connections. This means that connections can only run from neurons of a given layer to the next layer (never in the opposite direction). Therefore, such a network is sometimes called unidirectional topology and often takes the form of a three-layer model: input layer, hidden layer, and output layer (as the results show presented in Table 3). The number of neurons in the hidden layer implies the complexity of the generated network, which in fact represents the complexity of the issue being modelled. The existence of hidden layers allows the ANN to create models corresponding to the description of non-linear relationships.

The value of the created multi-contamination rank index is conceptional (original authors’ idea); therefore, there are no literature data corresponding to the values obtained in this study. Here, the highest values represent the lowest toxicity level, while the lowest values of this index represent the highest concentrations of the toxic and harmful elements. The performed analyses with the use of neural networks indicate that the overall accumulation of the harmful and toxic elements in post-hatching eggshells of grey herons was mainly influenced by urban fabrics, industrial areas, and pastures in their potential foraging areas (Table 4).

A high relative contribution of urban fabrics and industrial areas around heronries was associated with low CONT_IN levels. Some of the studied colonies are situated very close to the Upper Silesian conurbation with 2.1 million inhabitants (DD, ZB colonies) and to the Warsaw conurbation with 1.75 million inhabitants (RA colony) [35,36] and to other highly industrialized, urbanized, and densely populated areas with smelting plants, coal and metal mines, power plants, etc. being the source of contaminations of different types [37,38].

Concetrations of some studied elements were characterized by a relatively high standard deviation (see Appendix A). It can be explained by the fact that in some locations, one or a few eggshells contained a relatively higher concentration of a certain element, while this element was not found in other eggshells from the same location. This pattern can be interpreted by the opportunistic foraging of grey herons in various habitats, including rivers characterized by long-distance transport of contamination provided by nearby rivers [39,40,41]. Lower values of contaminants in eggshells in the present study as compared to data from the same species eggshells from the 1990s [3,35,41] may be interpreted in the context of decreasing industrial emissions to Polish rivers after economic transformations in Poland. However, the current elevated Pb concentrations (0.37–0.66 mg/kg) compared to the 1990s (0.05–0.30 mg/kg [3]) is alarming.

Modeling performed with the use of ANNs showed a significant role of pastures in lowering the concentrations of toxic elements as indicated by the increase in CONT_IN [40,41]. Grey herons forage in pastures to supplement their basic diet (fish) with invertebrates and mammals [10].

Our analyses show that the amount of Hg accumulated in grey heron eggshells was primarily influenced by the relative fraction of industrial areas, inland waters, and scrub vegetations (Table 4). In Europe, industrial and energy sectors serve as the main emitters of Hg into the environment [42,43]. Considering grey heron diet composition, aquatic prey, including fish, seems to be the main source of Hg for the studied grey heron eggshells. Surprisingly, low Hg levels were associated with a higher area of inland waters (lakes, reservoirs, and rivers) that can be explained by the fact that grey herons due to their anatomic limitations and foraging techniques may exploit only the shallow water zone of inland waters.

The ANNs revealed that the high level of Pb in eggshells was associated with a high contribution of relative area of permanent crops and a low share of heterogeneous agricultural areas in potential foraging areas of grey herons. Many studied heronries are located in river valleys characterized by the presence of orchards [44]. These habitats are characterized by an intense use of various agrochemicals, including insecticides and fungicides, serving as a source of Pb; moreover, orchards in river valleys are periodically contaminated with Pb from riverside water during floods [45,46]. Grey herons may hunt in this habitat type for small mammals and insects. The highest levels of Pb were found in colonies located on the largest Polish rivers—the Vistula (KR and RA) and Odra (KI)—that may indicate long-distance river transport of this element. This study revealed that a high relative area of heterogeneous agricultural areas around heronries was associated with low Pb accumulation in eggshells. This habitat type in Poland is mainly characterized by small allotments with limited use of agrochemicals [47,48,49] that may result in a lower level of Pb contamination.

The analysis of the results indicates that Cd concentrations are positively correlated with the degree of coverage of colony buffers with artificial, non-agricultural vegetated areas and permanent crops (Table 4). It suggests that agrochemicals, including pesticides and fertilizers characterized by a high Cd content, may serve as the main source of Cd [50,51]. The processes of Cd accumulation in the eggshells of the studied birds are best described by the RBF network (Table 3). This may indicate that the accumulation of this element can be more complex in nature (non-linear dependence) associated with the mosaic structure of crops in Poland combining intense and extensive farming with various use of agrochemicals [52,53].

The Mn concentration in heron eggshells was mostly influenced by the presence of industrial and marine areas. It is in concordance with the literature data showing that Mn compounds are emitted into the environment by factories, especially in metallurgic and chemical industries [42]. Mn is used in the chemical, textile, ceramic, and electrochemical industries and in the manufacturing of dyes, plant protection agents, and fertilizers. Car fumes are also an important source of Mn in the environment because of the widespread use of methylcyclopentadienyl manganese tricarbonyl (MMT) in gasoline to increase the octane rating [54,55]. Heronries with high average Mn concentrations in eggshells (mean >4.0 mg/kg; max >15.0 mg/kg) (MO, ZB, KI, RA, and OT) were located in highly industrialized and urbanized areas. Simultaneously, a relatively high Mn concentration was found in a nearby river sediments of Odra and Vistula Rivers (range 400–1737 mg/kg) [56,57]. The importance of marine habitats for Mn concentration in the grey heron eggshells may have resulted from a relatively high contribution of this habitat type in potential foraging areas, which were also characterized by a relatively high share of industrial and highly populated areas (MO) and/or they were situated close to estuaries of Odra (GA) and Vistula (KR) Rivers. The anatomy of the grey heron restricts its foraging ability to shallow water zones; therefore, in the DD heronry located nearby a water reservoir with an average depth of c.a. 15 m and a sediment Mn concentration of 817 mg/kg dw [58], we found only 1.34 mg/kg of Mn in the eggshells. In the heronry associated with the very shallow Otmuchów Lake (OT) (average depth of ca 7 m and sediment Mn concentration 86.4 mg/kg [59]), grey heron eggshells accumulated the highest amount of Mn among the studied colonies (6.04 mg/kg).

Urban fabrics and industrial areas in grey heron foraging buffers were the main factors influencing Fe concentrations in grey heron eggshells. Both habitat types are characterized by numerous sources of Fe emissions to the environment in the form of crustal and industrial dusts, sewages, or filtrates [60,61]. A relatively high Fe concentrations (>12 mg/kg) were found in colonies located close to industrial areas where large amounts of Fe were also found in the sediments of wetlands [56,59].

The As content in the eggshells of grey herons increased with increasing relative areas of urban fabrics, industrial areas, mines, and dumps and by the presence of construction sites (Table 4). As is widely used in the industrial sector for the production of wood preservatives, pharmaceuticals, alloying, glass, semiconductors, batteries, pigments, and metal adhesives [62]. In Poland, a significant source of As in the environment is also coal combustion [63] or the mining and processing of copper ore [64,65]. Among the studied colonies, those located in SW Poland (IS, MA, WR, ST, and OT) were characterized by the highest As concentrations in eggshells. Elevated concentrations of As were observed in the eggshells from colonies associated with the presence of mining and Cu processing industries (JS and MA). The MA colony was characterized by the highest average As level among all studied heronries. In the case of the WR, ST, and OT colonies, a relatively high contribution of urbanized and industrial areas in the foraging grounds favored the accumulation of this element.

The colony in WR is located in the city zoological garden located in the core part of Wrocław city with 640,000 inhabitants. However, the amounts of As found in grey heron eggshells from this colony were within its environmental background ranges (0–3 mg/kg dw) [63]. This corresponds to the previously observed low ability of As to enter ecosystem pathways [66,67].

## 5. Limitations of the Study

Our study on modeling elemental concentration in grey heron eggshells, like other studies based on a restricted set of data, has some limitations. The present study was focused on the differences in elemental concentrations in avian eggshells in the context of type of land use in 20 km buffers around colonies. The detailed studies of elemental concentrations in potential food or in the nearest environment were not carried out. The role of other factors, such as metabolic state and health in the process of sequestration, was also omitted/either not investigated?. As non-essential metals are sequestered into the eggshell for excretion, the post-hatched eggshells, which are easy to collect in colonies, may provide a convenient, non-invasive tool for monitoring (*sensu stricto* assessing avian exposure to habitat contamination) heavy metal contaminations in water birds. Considering the opportunistic foraging of grey herons, the contamination level in eggshells is representative not only of the aquatic ecosystems but of all habitats within the foraging range.

Despite the mentioned limitations, the analyses of the contamination levels in the post-hatched eggshells non-invasively collected in grey herons may serve as a proper tool for monitoring trace element contamination in birds. In areas where the grey heron does not occur or is a rare breeder, eggshells may be collected from other top predatory waterbirds. The colonially breeding species are the most recommended biomonitors because one may relatively easily collect a considerable sample size. Of course, conservation status and susceptibility to short-term disturbance (visit in the colony to collect eggshells if done before the end of the breeding period) should be considered before making the choice of species.

## 6. Conclusions

Our analysis shows that ANN may be used effectively to predict a link between the concentration of toxic elements in post-hatched eggshells of grey herons and the habitat composition in the potential foraging areas of this bird. The proposed ANN may be used at least in the areas with similar types of main pollutants present in the environment. In the case of highly industrialized and/or polluted areas, ANN should be adjusted for local conditions. However, after such an adjustment, it can be used as a practical tool to help with monitoring and management.

In the presented research, we show that the content of elements in grey heron eggshells is associated with the relative area of habitat types in potential foraging grounds. The analysis shows that relative areas of urban fabrics, industrial areas, pastures, and inland waters were strongly associated with the presence of harmful elements; the role of forests proved to be negligible. We did not detect any link between the contamination and geographic location, which makes the model more universal. The presented results are important from an ecological perspective.

## Figures and Tables

**Figure 1 sensors-22-03723-f001:**
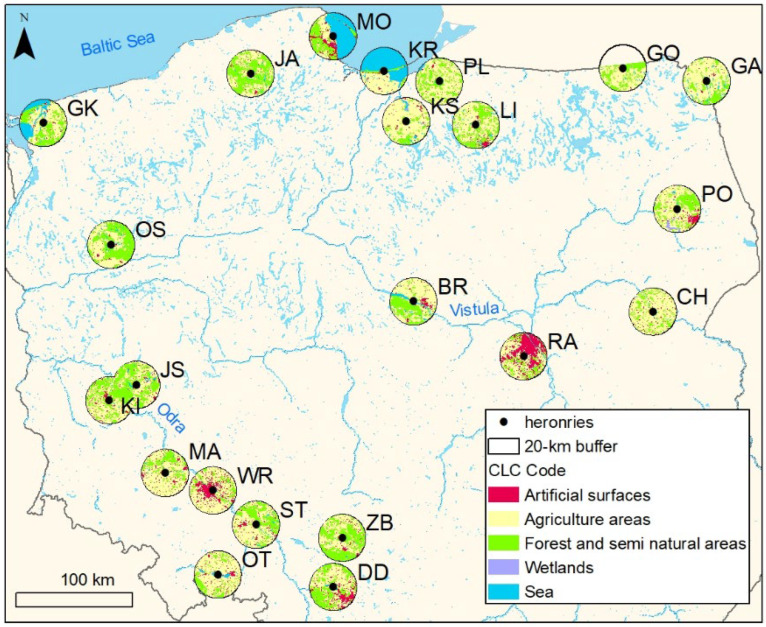
Study area showing locations of the investigated grey heron colonies and main habitat types according to the Corine Land Cover CLC2012 model, level 1; (https://land.copernicus.eu/pan-european/corine-land-cover/clc-2012 (accessed on 1 February 2022)) in 20 km buffers around the colonies representing potential foraging areas.

**Figure 2 sensors-22-03723-f002:**
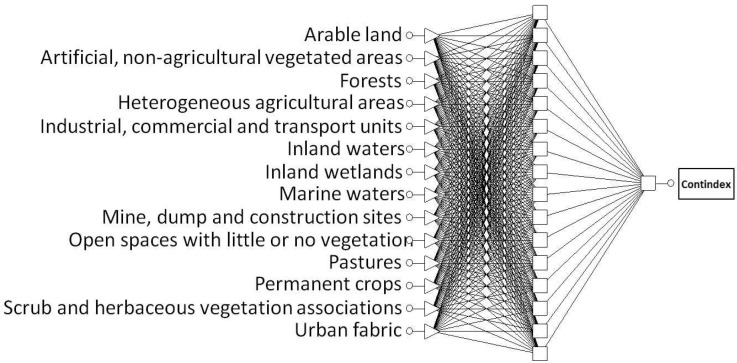
Optimal structure of ANN type MLP: 14-16-1 for CONT_IN.

**Table 1 sensors-22-03723-t001:** Input and output variables of ANN.

	**Input (Descriptors) *^,#^**	**Designation**
	**Variable Name/Type of Land Use**	
1	Arable land	ARA_LAND
2	Artificial, non-agricultural vegetated areas	ARTIF_AREAS
3	Forests	FORESTS
4	Heterogeneous agricultural areas	HET_AGRO
5	Industrial, commercial, and transport units	INDUST_UNITS
6	Inland waters	INL WAT
7	Inland wetlands	INL_WET
8	Marine waters	MAR_WAT
9	Mines, dumps, and construction sites	MINE_DUMP
10	Open spaces with little or no vegetation	OPEN_SPACE
11	Pastures	PASTURES
12	Permanent crops	PERM_CROPS
13	Scrub and/or herbaceous vegetation associations	SCRUB_VEGET
14	Urban fabric	URBAN_FABRIC
	**Output** **Variable name (Bioindicators) ^#^**	**Designation**
A	Contamination index **	CONT_IN
B	Mean Hg(mg/kg)	Hg
C	Mean Pb (mg/kg)	Pb
D	Mean Cd (μg/kg)	Cd
E	Mean Fe (mg/kg)	Fe
F	Mean Mn (mg/kg)	Mn
G	Mean Zn (mg/kg)	Zn
H	Mean As (mg/kg)	As

* Dimensionless data representing the relative rate of various habitat types in potential foraging areas. ** dimensionless. ^#^ values without dimensions were considered in the model.

**Table 2 sensors-22-03723-t002:** Values of multi-contamination rank index (CONT_IN) calculated based on the measured concentrations of elements (number of eggshells available per location in brackets). Note: The lowest values indicate the highest sum of the mean concentrations of the harmful or toxic elements (Al, As, Cd, Cr, Cu, Fe, Hg, Mn, Se, Sr, V, and Zn). Colony codes—see Section 2.1.

Colony Code	CONT_IN	Colony Code	CONT_IN
GK	237	DD	247
KR	128	JA	102
BR	156	PO	177
CH	168	MA	134
KS	169	LI	167
GA	262	KI	120
GO	232	RA	170
OS	254	OT	144
JS	280	ST	157
MO	195	PL	188
ZB	197	WR	163

**Table 3 sensors-22-03723-t003:** Regressions statistics of the obtained optimal neural models. S.D. ratio—quotient of standard deviations determined for errors and for data. Correlation—standard R-Pearson correlation coefficient between the results given by the generated neural model and the actual output values.

	Learning File	Validation File	Test File	Type of Neural Model
		**CONT_IN**		
S.D. ratio	0.0718400	0.0190262	0.6633987	MLP: 14-16-1
Correlation	0.9975369	0.9998225	0.9869675	
		**Toxic elements**		
		**Hg**		
S.D. ratio	0.06611	0.0745	0.19449	MLP: 14-3-1
Correlation	0.9981396	0.9974322	0.9822994	
		**Pb**		
S.D. ratio	0.06047	0.03304	0.06773	MLP: 14-8-1
Correlation	0.9120701	0.8037518	0.972723	
		**Cd**		
S.D. ratio	0.004494	0.005565	0.006037	RBF: 14-8-1
Correlation	0.7052663	0.8266247	0.7990223	
		**Essential elements**		
		**Fe**		
S.D. ratio	0.1003075	0.277002	0.2561513	MLP: 14-3-1
Correlation	0.9951584	0.886153	0.8936853	
		**Mn**		
S.D. ratio	0.1890534	0.2848873	0.2264988	MLP: 14-3-1
Correlation	0.9821074	0.8758581	0.8571459	
		**Zn**		
S.D. ratio	0.253584	0.2057967	0.1674033	MLP: 14-47-1
Correlation	0.7570912	0.7174649	0.6284907	
		**As**		
S.D. ratio	0.1137984	0.4906985	0.526818	MLP: 14-25-1
Correlation	0.9939374	0.9460305	0.8500785	

**Table 4 sensors-22-03723-t004:** The sensitivity analysis for the most important input variables (in italics) for the examined output variables (in bold) and Pearson correlation coefficient (r) for these variables.

Output ANN Variable	Rank of Input Variables in ANN		
	1	2	3
**CONT_IN**	*URBAN_FABRIC*	*INDUST_UNITS*	*PASTURES*
Error	61.93411	55.98893	50.2746
Ratio	16.40922	14.83407	13.32008
r	−0.116	−0.025	0.347
Toxic elements			
**Hg**	*INDUST_UNITS*	*INL_WAT*	*SCRUB_VEGET*
Error	0.06141	0.0583	0.0544
Ratio	12.86131	12.20809	11.39253
r	0.029	−0.079	0.075
**Pb**	*PERM_CROPS*	*HET_AGRO*	*SCRUB_VEGET*
Error	0.1144253	0.1116625	0.1056726
Ratio	1.165484	1.137344	1.076334
r	0.153	−0.294	0.025
**Cd**	*ARTIF_AREAS*	*PERM_CROPS*	*ARA_LAND*
Error	0.006896	0.006581	0.006522
Ratio	1.133515	1.081717	1.071923
r	0.121	0.094	−0.009
Essential elements			
**Fe**	*URBAN_FABRIC*	*INDUST_UNITS*	*PASTURES*
Error	7.788201	6.93745	3.479581
Ratio	19.19456	17.09783	8.575668
r	0.102	0.022	−0.376
**Mn**	*INDUST_UNITS*	*MAR_WAT*	*PERM_CROPS*
Error	7.960049	5.096137	5.093208
Ratio	25.89732	16.57984	16.57031
r	0.027	0.018	−0.3634
**Zn**	*FORESTS*	*ARTIF__AREA*	*SCRUB_VEGET*
Error	9.796726	9.366866	8.412936
Ratio	1.19883	1.146227	1.029494
r	0.337	0.193	0.161
**As**	*URBAN_FABRIC*	*INDUST_UNITS*	*MINE_DUMP*
Error	0.3518201	0.2070515	0.2014431
Ratio	12.84457	7.559226	7.354468
r	0.261	0.175	0.211

Note: Rank indicates the significance level of the input variable and orders the variables according to importance: number 1 means the dominant variable and orders the variables by importance (by decreasing error); error indicates the quality of the network in the absence of a given variable: the lower the rank number of the ANN input variable, the lower the rank number of the input variable ANN, the larger the error made by the network without this variable; ratio—the ratio of the network reduced error by the SSN error obtained using all the variables; if the quotient is lower than 1.0, removing the variable improves the ANN quality. Habitat types—see Table 2.

## Data Availability

Data available on request.

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
