# Peer review of "Identification of Factors Affecting Environmental Contamination Represented by Post-Hatching Eggshells of a Common Colonial Waterbird with Usage of Artificial Neural Networks"

_sensors, 2022, doi:10.3390/s22103723_

Round 1

Reviewer 1 Report

Dear Authors,

I have finished my review on your paper. I think that it is interesting and would worth publishing. Nevertheless, it requires significant review before that. Please refer to my following comments.

BR,

R.

PAPER IN GENERAL: the topics/approach is of interest for the general science and could add value to the Sensors journal. However, I’ve seen few things in this paper dealing with sensors. Excepting one sentence in which the authors try to justify that the eggshells may act as sensors (which I doubt – more as biological data storing media), there is less information about this topic. However, I find this approach as valuable and the paper could be published in Sensors following a careful revision and augmentation of ideas so as to qualify it for the journals. Please refer to my following comments.

TITLE: Rephrase it to exactly reflect what has been done and to be understandable for the audience of the journal.

ABSTRACT:

General comment: Introduction is too abrupt. Complement the need of your study in 1-2 sentences. Do not use names/variables of your experiment without properly introducing them (i.e., Urban fabric, Industrial). Refer to them in more general terms. Refocus the results given in short after improving the paper. Conclude in 1-2 sentences.

KEYWORDS: OK.

INTRODUCTION:

General comment:

In general, well written and informative, although English phrasing needs improvement. However, there is a need for a revision here in particular to make the concepts used more clearer and to argument the choice of methods.

Specific comments:

Lines 36-37: this sentence does not belong here. It is about methods.

Line 39: be more specific so as to indicate that the diet is dominated.

Line 40: check the syntax.

Line 41: essential – I am not sure if it the right word. Maybe micro?

Line 45: what do the authors mean by “local”? Be more specific. How wide could be the area?

Line 46: be more specific. What elements?

Lines 47-49: need rephrasing.

Line 49: same as above.

Lines 51-55: same as above in terms of area. Be more specific. How wide? Then again, the sensor is a concept used to gather data not to store it. Perhaps the concept of signal would be better fit here.

Lines 55-59: some parts are redundant as the diet was presented at the beginning of intro. Merge the info in one sentence/paragraph.

Lines 60-64: this is the information about the “sensing range”. Since this is important, merge it in a single paragraph with all the references given above. See comments from L45, 51-55.

Lines 64-66: I am not sure if unpublished information can be given as a reference.

Lines 67-71: I am not sure where the authors are going with these sentences. Maybe a rephrasing would make it clearer.

Lines 71-72: I am not sure if the sentences from above make a good reason to use ANN. Here is a weak point of the introduction because it does not specify why the use of ANN is important.

Lines 74-80: I agree with the statements, however, there seems that the references are biased towards the Poland. Another problem is that not only ANN make a suitable tool for classification and regression of multidimensional nonlinear data. Many other machine learning algorithms are used for similar purposes. For instance, RF and SVM. Therefore, I would suggest to document better the reasons of choosing/using ANNs.

MATERIALS AND METHODS:

General comment:

In general, well written and described. However, the main issue is the ANN for which the architecture, activation functions, performance metrics selected to check the training and generalization capability, and so on, need to better described. Otherwise, the experiment is difficult to understand.

Specific comments:

Figure 1: very nice figure. To enhance its readability, I would recommend to augment the circles by placing them on the sizes with arrows pointed towards their origin and to enhance their scale.

Lines 112-119: place this in regular text or in the figure legend.

Lines 129-136: is there any reference data on what non-contamined would be? Such data would be valuable to rank the different degrees of contamination.

Lines 138-152: how about the potential contaminators? And about the diet…Is there any reference data on how the migratory populations used as food may affect the results?

Lines 160-162: I assume that there was a horizontal stacking of the input data? Also, rephrase for clarity.

Lines 162-163: based on what?

Line 165: by what technique? Min-max? Normalization?

Table 2: give also the units.

Lines 176-186: More specifics are needed here. Also, if your output layer was categorical (I failed to understand what kind of data it contained) then perhaps other performance metrics would be appropriate. If it was numerical and your ANN was developed for regression-prediction, then maybe you could use some error metrics. How good did it performed? Was it able to learn and generalize well? How were these measured?

RESULTS:

General comment:

In general, well written. However, some improvements are needed here. Parts that should be explained beforehand need to be addressed in the Materials and methods since the authors are using them here to explain their results. Some data adds no value to the results and should be excluded. Some needs to be added.

Specific comments:

Lines 188-190: remove this.

Line 191: check the font type of this subheading.

Lines 192-209: Add a table and give the figures in it, including that of supplementary materials to which I did not had access. This is an important part of your results as they are fed in ANN.

Figure 2: adds no value to your paper since you can say in one line how many hidden neurons were there. If the weights are not present, that’s the case…

Lines 241-244: these concepts need to be described beforehand in Materials and methods. Here the reader will have difficulty to understand them.

Table 4: I see no utility for it here. Particularly since it does not give hints for what the data stands in columns. Are there probabilities?

Lines 251-254: these concepts need to be described beforehand in Materials and methods. Here the reader will have difficulty to understand them.

Line 259: equipped – holding a neuron.

Lines 258-265: are these the findings of the authors or were they documented before?

DISCUSSION:

General comment:

Too long and less interpretative in the view of findings. The paper is about the ANN and is sent to Sensors Journal. So, the readers would expect to see something more about the application of ANN to your problem and less about the mechanisms of pollutants transportation. In my opinion, it needs to be rewritten. Are there any limitations in your approach?

Specific comment:

Lines 326-327: I failed to see where the geographical location was contextualized.

Lines 359-360: see my comments from introduction.

All paragraphs: why so many emphasis to explaining mechanisms of pollutants since the paper is about the use of ANN. I would have been expected to see here interpretations of the models develop and tested. These 5 paragraphs can be largely reduced to a single one.

CONCLUSION:

General comment:

Most text is out of context. Needs rephrasing/rewiring.

Specific comment:

Lines 477-478: But you said that it is location invariant in some part of the text before...And you say in the next paragraph that your model is universal… Perhaps some data augmentation techniques could solve this.

LITERATURE/CITING SYSTEM:

General comment:

At a first glance, more than half of the studies are coming from Polish authors/references. Most of them are out of context judging by the fact that introduction, materials and methods and discussion need significant revision. Perhaps the authors would find more value in searching for papers dealing specifically with ANNs. There are lots published by MDPI.

Author Response

Answers to Reviewer 1

We thank the Reviewer for useful comments and remarks which helped us to improve the quality of our work. Please find below the detailed answers to the questions which were raised during a review. All the corrections are marked in yellow in the manuscript text. We hope that in the present form our work is suitable to be published in Sensors MDPI special issue: Identification of Bio- and Eco-Materials Using Advanced Computational Methods.

PAPER IN GENERAL: the topics/approach is of interest for the general science and could add value to the Sensors journal. However, I’ve seen few things in this paper dealing with sensors. Excepting one sentence in which the authors try to justify that the eggshells may act as sensors (which I doubt – more as biological data storing media), there is less information about this topic. However, I find this approach as valuable and the paper could be published in Sensors following a careful revision and augmentation of ideas so as to qualify it for the journals. Please refer to my following comments.

We agree with the Reviewer that eggshells of a grey heron is a bio-material which is here used as a biological data storing media. This information has been added (lines 21-22 and 37). We also explained in more detail why we used Artificial Neural Networks (lines 72-87).

TITLE: Rephrase it to exactly reflect what has been done and to be understandable for the audience of the journal.

The title has been changed to: Identification of factors affecting environmental contamination represented by post-hatching eggshells of a common colonial waterbird with usage of artificial neural networks

ABSTRACT: General comment: Introduction is too abrupt. Complement the need of your study in 1-2 sentences. Do not use names/variables of your experiment without properly introducing them (i.e., Urban fabric, Industrial). Refer to them in more general terms. Refocus the results given in short after improving the paper. Conclude in 1-2 sentences.

This part of the manuscript has been written anew, see lines 18-33.  

KEYWORDS: OK.

INTRODUCTION:General comment: In general, well written and informative, although English phrasing needs improvement. However, there is a need for a revision here in particular to make the concepts used more clearer and to argument the choice of methods.

Specific comments:

Lines 36-37: this sentence does not belong here. It is about methods.

The sentence from lines 36-37 has been removed. Relevant information has been given in Materials and methods section.

Line 39: be more specific so as to indicate that the diet is dominated.

REPLY: The information of a diet is already placed in the introduction. “It is a highly opportunistic predator, with a diet dominated by fish, crustaceans or mammals depending on habitat and season; depending on location, it may be [8, 9].” See lines:

Line 40: check the syntax.; Lines 47-49: need rephrasing.

REPLY: The syntax of the above sentences has been checked and parts of the sentences re-phrased.

Line 41: essential – I am not sure if it the right word. Maybe micro?

REPLY: Yes we agree with the Reviewer. It should be toxic and harmful elements.

Line 45: what do the authors mean by “local”? Be more specific. How wide could be the area?

REPLY: We added the specific information that the area means vicinity of colony where birds breed (lines 26 and 47)

Line 46: be more specific. What elements?

REPLY: toxic and non-toxic elements.

Lines 51-55: same as above in terms of area. Be more specific. How wide? Then again, the sensor is a concept used to gather data not to store it. Perhaps the concept of signal would be better fit here.

REPLY: We have added a concept of a colonial waterbird, the grey heron Ardea cinerea usage as a biological data storing media for biomonitoring. As it breeds up to 20 km from the colony, this distance has been referred as a sensing range (see lines 21-22, 36-37 and 55-56)

Lines 55-59: some parts are redundant as the diet was presented at the beginning of intro. Merge the info in one sentence/paragraph.

REPLY: Sentences with similar meanings that are found in different parts of the Introduction have been grouped into one paragraph

Lines 60-64: this is the information about the “sensing range”. Since this is important, merge it in a single paragraph with all the references given above. See comments from L45, 51-55.

REPLY: The information about “ sensing range” was added. We are grateful for that useful remark.

Lines 64-66: I am not sure if unpublished information can be given as a reference.

REPLY: We have corrected it accordingly, by adding a reference from other authors [10].

Lines 67-71: I am not sure where the authors are going with these sentences. Maybe a rephrasing would make it clearer.

REPLY: As there were some other remarks concerning earlier parts of the text the manuscript,  text has been thoroughly re-phrased. See lines:

Lines 71-72: I am not sure if the sentences from above make a good reason to use ANN. Here is a weak point of the introduction because it does not specify why the use of ANN is important.

We have better justified the usage of ANNs , see lines 59-87.

We also added references containing the latest research published in MDPI

Shi, T.; Liu, H.; Chen, Y.; Fei, T.; Wang, J.; Wu, G. Spectroscopic Diagnosis of Arsenic Contamination in Agricultural Soils. Sensors2017, 17, 1036. https://doi.org/10.3390/s17051036

Bieganowski, A.; Józefaciuk, G.; Bandura, L.; Guz, Ł.; Łagód, G.; Franus, W. Evaluation of Hydrocarbon Soil Pollution Using E-Nose. Sensors 2018, 18, 2463. https://doi.org/10.3390/s18082463

Fuentes, S.; Gonzalez Viejo, C.; Tongson, E.; Lipovetzky, N.; Dunshea, F.R. Biometric Physiological Responses from Dairy Cows Measured by Visible Remote Sensing Are Good Predictors of Milk Productivity and Quality through Artificial Intelligence. Sensors2021, 21, 6844. https://doi.org/10.3390/s21206844

Dominguez-Morales, J.P.; Duran-Lopez, L.; Gutierrez-Galan, D.; Rios-Navarro, A.; Linares-Barranco, A.; Jimenez-Fernandez, A. Wildlife Monitoring on the Edge: A Performance Evaluation of Embedded Neural Networks on Microcontrollers for Animal Behavior Classification. Sensors2021, 21, 2975. https://doi.org/10.3390/s21092975

Fuentes, S.; Tongson, E.; Unnithan, R.R.; Gonzalez Viejo, C. Early Detection of Aphid Infestation and Insect-Plant Interaction Assessment in Wheat Using a Low-Cost Electronic Nose (E-Nose), Near-Infrared Spectroscopy and Machine Learning Modeling. Sensors 2021, 21, 5948. https://doi.org/10.3390/s21175948

Lines 74-80: I agree with the statements, however, there seems that the references are biased towards the Poland. Another problem is that not only ANN make a suitable tool for classification and regression of multidimensional nonlinear data. Many other machine learning algorithms are used for similar purposes. For instance, RF and SVM. Therefore, I would suggest to document better the reasons of choosing/using ANNs.

REPLY: We agree with the Reviewer that other algorithms of binary regression instruments, dedicated to solving various prediction problems, such as: decision trees, SVM models, rules (IF...THEN...), etc., are well known and sometimes used. The choice of neural prediction techniques was dictated, among others, by the fact of their high performance (quality of generated forecasts), versatility, low cost and ease of use. Due to their numerous advantages SSNs (e.g. as predictive tools) are now widely used in scientific and utilitarian applications.

MATERIALS AND METHODS:General comment: In general, well written and described. However, the main issue is the ANN for which the architecture, activation functions, performance metrics selected to check the training and generalization capability, and so on, need to better described. Otherwise, the experiment is difficult to understand.

Specific comments:Figure 1: very nice figure. To enhance its readability, I would recommend to augment the circles by placing them on the sizes with arrows pointed towards their origin and to enhance their scale.Lines 112-119: place this in regular text or in the figure legend.

REPLY: Figure 1 was generated using a Corine Land Cover CLC2012 model. We did our best to make the increase of readability of this figure. We also shifted the lines 112-119 describing grey heron colonies codesto the regular text.

Lines 129-136: is there any reference data on what non-contamined would be? Such data would be valuable to rank the different degrees of contamination.

No data exists on the level of toxicity of the elements based on their concentration in eggshells. To illustrate levels of toxic elements in eggshell from particular colony in one index we proposed  multi-contamination rank index. The value of the created multi-contamination rank index is conceptional (original Authors’ idea) therefore there are no literature data corresponding to the values obtained in this study. Here the highest values represent the lowest toxic level while lowest values of this index represent the highest concentrations of the toxic and harmful elements.  In the extreme the example calculation shows that  the theoretically most polluted environment it will take the value of 22 and for the least polluted 484. As seen from our data its value ranges between 102 and 280.

Lines 138-152: how about the potential contaminators? And about the diet…Is there any reference data on how the migratory populations used as food may affect the results?

REPLY: We have referred to potential sources of contamination in Discussion (industry, agrochemical etc.). regarding migratory prey – the grey heron mainly forage on local non-migratory organisms as fish, insects or small mammals. Individuals foraging in rivers may prey on some migratory fish species but we think that that effect is negligible.

Lines 160-162: I assume that there was a horizontal stacking of the input data? Also, rephrase for clarity.

The structure of the learning set was represented by an empirical data set containing 44 measurement cases. Each learning case included 14 input variables (descriptors) and 1 variable, output, relevant to the predicted variable. The collection was subdivided in a standard way into 3 subsets: training containing 33 cases, validation containing 5 cases, and test containing 6 cases. Non-symmetrical division of training cases (33:5:6) was justified by the limited number of empirical data (in relation to the number of descriptors), which resulted, among others, from the high cost of conducted research.

Lines 162-163: based on what?Line 165: by what technique? Min-max? Normalization?

The training, validation, and test cases that form part of the learning set have been subjected to standard pre-processing procedures (implemented in the STATISTICA package in the ST Neural Networks module), involving scaling their values to normalized numerical intervals. The applied pre- and post-processing techniques and the parameters controlling their operation are stored in the ST Neural Networks program in a file containing the definition of SSN. The Pre- and Post-Processing tool available in the SSN Editor was used to create neural models. The Minimax function (without data normalization) was used as the conversion function.

Table 2: give also the units.

Units of elements concentrations have been added, while some ????? are dimensionless. In the model values were used.

Lines 176-186: More specifics are needed here. Also, if your output layer was categorical (I failed to understand what kind of data it contained) then perhaps other performance metrics would be appropriate. If it was numerical and your ANN was developed for regression-prediction, then maybe you could use some error metrics. How good did it performed? Was it able to learn and generalize well? How were these measured?

The output layer of generated SSNs consisted of one neuron each (example shown in Fig. 2 for the predicted variable CONT-IN) representing individual bioindicators included in Table 2 (Output Variable name (Bioindicators)). The results of the simulations carried out and the results of the sensitivity analysis of the generated ANNs to the individual input variables are shown in Tables 5 and 6.

RESULTS:General comment: In general, well written. However, some improvements are needed here. Parts that should be explained beforehand need to be addressed in the Materials and methods since the authors are using them here to explain their results. Some data adds no value to the results and should be excluded. Some needs to be added.

Specific comments:

Lines 188-190: remove this.

REPLY: Unneeded text has been removed.

Line 191: check the font type of this subheading.

REPLY: Font in the sub-heading has been changed.

Lines 192-209: Add a table and give the figures in it, including that of supplementary materials to which I did not had access. This is an important part of your results as they are fed in ANN.

REPLY: We decided to leave the tables containing results as a Supplementary files (4 full pages of data). We do not know how this happened but other Reviewer easily reached this data. Data will be available on request.

Figure 2: adds no value to your paper since you can say in one line how many hidden neurons were there. If the weights are not present, that’s the case…

REPLY: We believe that figure 2 is an important example. It helps to understand how ANNs were created. It has been confirmed by the above question about horizontal stacking of the data.

Lines 241-244: these concepts need to be described beforehand in Materials and methods. Here the reader will have difficulty to understand them.

The following explanation has been added to Materials and Methods:

The generated topologies of the multilayer perceptron type (MLP with 1 hidden layer) were hybrid learned by the BP (Back Propagation) algorithm and then learned by the CG (Conjugate Gradients) algorithm. The generated network with radial basis functions (RBF type) was learned in 3 stages: KM (K-Means: K-means), KNN (K-Nearest Neighbor: K-nearest neighbors), and PI (pseudo-inversion, linear least squares optimization) methods.

Table 4: I see no utility for it here. Particularly since it does not give hints for what the data stands in columns. Are there probabilities?

Table 4 has been removed from the manuscript

Lines 251-254: these concepts need to be described beforehand in Materials and methods. Here the reader will have difficulty to understand them.

REPLY:Additional information was added, see lines 184-211.

Line 259: equipped – holding a neuron. REPLY: Yes we changed it accordingly.

Lines 258-265: are these the findings of the authors or were they documented before?

REPLY: Information has been added , see lines 273-281.

RBF neural network had 1 hidden layer made of 8 radial neurons and 1 output layer holding a linear neuron. Such a network is characterized by a short learning time and good quality of operation, but at the cost of a high complexity of the structure. Networks with radial base functions learn relatively quickly and have the advantage that they never extrapolate functions too far from known data. So they are in a way the safest. However, as a rule, they are much larger than MLP networks that solve the same problems. This fact makes their operation on the ANN simulating computer more time-consuming.

DISCUSSION: General comment:  Too long and less interpretative in the view of findings. The paper is about the ANN and is sent to Sensors Journal. So, the readers would expect to see something more about the application of ANN to your problem and less about the mechanisms of pollutants transportation. In my opinion, it needs to be rewritten. Are there any limitations in your approach?

Specific comment:

Lines 326-327: I failed to see where the geographical location was contextualized.

Neural models preliminary created with consideration of geographic location did not show dependence of concentrations of particular elements on this parameter. The sensitivity analysis of the individual networks showed that the rank of the location descriptor is 15, showing that this variable does not affect the quality of the obtained networks. Therefore, this variable was removed from the model. We concluded that if this variable is not necessary to the model than it becomes location-independent, thus the issue can be considered based on the characteristics of the types of land use; for details see lines 350 -363

We thank the Reviewer for this useful remark.

Lines 359-360: see my comments from introduction.

REPLY: We have changed it adequately See lines 402-403: “Grey herons forage in pastures to supplement their basic diet (fish) with invertebrates and mammals [10]”.

All paragraphs: why so many emphasis to explaining mechanisms of pollutants since the paper is about the use of ANN.I would have been expected to see here interpretations of the models develop and tested. These 5 paragraphs can be largely reduced to a single one.

The subject of Discussion section has been largely reduced concerning explanations of the mechanisms of pollution. Instead  part of the text has been added explaining the results obtained with use of ANNs.

CONCLUSION:General comment:

Most text is out of context. Needs rephrasing/rewiring. Specific comment:

Lines 477-478: But you said that it is location invariant in some part of the text before...And you say in the next paragraph that your model is universal… Perhaps some data augmentation techniques could solve this. ??????

We agree with the Reviewer and  therefore we added some explanations.

Thus, ANN may serve as an effective tool for prediction of contamination level of waterbirds tissues based on relative area of particular habitat types in the vicinity of colonies, at least in the areas with similar types of main pollutants present in the environment. In the case of highly industrialized and/or polluted areas ANN should be adjusted for local conditions; lines 509-510.

 LITERATURE/CITING SYSTEM: At a first glance, more than half of the studies are coming from Polish authors/references. Most of them are out of context judging by the fact that introduction, materials and methods and discussion need significant revision. Perhaps the authors would find more value in searching for papers dealing specifically with ANNs. There are lots published by MDPI.

The publication was supplemented with literature entries on applications of neural networks. We searched especially for those published by MDPI. We added

Shi, T.; Liu, H.; Chen, Y.; Fei, T.; Wang, J.; Wu, G. Spectroscopic Diagnosis of Arsenic Contamination in Agricultural Soils. Sensors2017, 17, 1036. https://doi.org/10.3390/s17051036

Bieganowski, A.; Józefaciuk, G.; Bandura, L.; Guz, Ł.; Łagód, G.; Franus, W. Evaluation of Hydrocarbon Soil Pollution Using E-Nose. Sensors2018, 18, 2463. https://doi.org/10.3390/s18082463

Fuentes, S.; Gonzalez Viejo, C.; Tongson, E.; Lipovetzky, N.; Dunshea, F.R. Biometric Physiological Responses from Dairy Cows Measured by Visible Remote Sensing Are Good Predictors of Milk Productivity and Quality through Artificial Intelligence. Sensors2021, 21, 6844. https://doi.org/10.3390/s21206844

Dominguez-Morales, J.P.; Duran-Lopez, L.; Gutierrez-Galan, D.; Rios-Navarro, A.; Linares-Barranco, A.; Jimenez-Fernandez, A. Wildlife Monitoring on the Edge: A Performance Evaluation of Embedded Neural Networks on Microcontrollers for Animal Behavior Classification. Sensors2021, 21, 2975. https://doi.org/10.3390/s21092975

Fuentes, S.; Tongson, E.; Unnithan, R.R.; Gonzalez Viejo, C. Early Detection of Aphid Infestation and Insect-Plant Interaction Assessment in Wheat Using a Low-Cost Electronic Nose (E-Nose), Near-Infrared Spectroscopy and Machine Learning Modeling. Sensors2021, 21, 5948. https://doi.org/10.3390/s21175948

Publications from Polish authors have been reduced to the necessary minimum.

Answers to Reviewer 2

We thank the Reviewer for useful comments and remarks which helped us to improve the quality of our work. Please find below the detailed answers to the questions which were raised during a review. All the corrections are marked in yellow in the manuscript text. We hope that in the present form our work is suitable to be published in Sensors MDPI special issue: Identification of Bio- and Eco-Materials Using Advanced Computational Methods.

The idea to focus on a top predator (in freshwater ecosystems), such as the grey heron, to monitor environmental changes in essential elements, pesticides and various pollutants is great. It is also clear that the grey heron adopts the income breeder strategy for acquiring nutrients for egg production and so their maternal investments (including eggshells) should correspond to the contamination of the local breeding environment.

We thank very much to the Reviewer for a positive assessment of our idea of usage of a grey heron eggshells to monitor the environment. Most of the questions asked by the Reviewer concern limitations of the study.

Please find below the detailed answers to the questions. We have added a part titled Limitations of the studies to the manuscript where the answers are included.

However, I have some questions that I think they should be answered:

  • The idea to use grey heron Ardea cinerea eggshells is great but what about the geographic areas where grey heron eggshells cannot be found? Is current study designed only for Poland?

REPLY: We used here the grey heron as a model species given its common occurrence in Poland. However, other top predators from freshwater ecosystems that can bio-magnify and bio-accumulate toxic and essential elements can be used as well. Herons, in that the grey heron have wide geographical range breeding in all regions except from polar regions. We have added one sentence about this issue to the part of Discussion about limitations of the study. See lines x-y: “In areas where the grey heron does not occur or is rare breeder, eggshells may be collected from other top predator waterbirds; the colonially breeding species are the most recommended because one may relatively easily collect considerable sample size.  

Although the study has been executed in Poland there is a possibility to broaden it to other geographic areas and other species being income breeders. The problem is however samples collection as well as costly elemental analyses. We would be delighted to broaden our research.

  • Since grey herons usually travel between 1-10 km from the nesting site (and even up to 20km if attractive food is found at a greater distance), how safe is to use eggshells as biological indices? The contamination in this case may represent a location up to 20km away.

REPLY: This 20 km range represents quite reasonable indication distance compared to the smaller species of birds. Passerines during the breeding season hold small territories, foraging within several hundred meters around the nest.

  • Obviously, herons focus on getting the food needed and they do not make choices, meaning they can get food from either a contaminated wetland or from a location near to industrial or highly populated areas. Therefore, it is difficult to deduce conclusions whether a near wetland is facing a pollution issue or not. As a result, these insights based on eggshells analysis seems unsafe to be used to determine the health of adjusting aquatic ecosystems. The sources of pollution due to human activities are already known in most cases, so what is the additional benefit of analyzing the heron eggshells?

REPLY: A signal from post-hatching eggshells represents contamination in all habitats explored by the herons. In the majority of cases they forage in aquatic ecosystems. In highly industrialized areas aquatic ecosystems are often contamination. Thus, relative area of various habitats can serve as a good predictor of all grey herons foraging habitats (not only aquatic) contamination and in turn contamination of their tissues and/or eggs.We have added one sentence to the part of Discussion about limitations of the study. See lines x-y: “Considering opportunistic foraging of grey herons, contamination level in eggshells are representative not only for aquatic ecosystems but all habitats within the foraging range”.

Knowledge of pollution level from human activities is not always monitored in the correct way and even if it is not always all data about it are easily available. In many cases ecotoxicological studies served as the first warning about high contamination of particular area (e.g. history of massive DDT use).

  • What is your proposal for monitoring changes over a region of interest based on heron eggshells (collecting eggshells in different intervals)? What if eggshells are not available at a given time? Is the population of herons (or their nests) under monitoring (is it stable over time)? I think you should add a limitation section.

REPLY:A general definition of monitoring or bio-monitoring says that it is an assessment of exposure to environmental and industrial chemicals by measuring the amounts of the chemicals or their metabolites (biomarkers) in tissues or fluidsand does notinclude tracing changes in time. And we use this word in this original meaning. Thus, we did not mention about possible changes in the indicator population size. But as we mentioned before, the grey heron is not the only species serving as a good indicator of  surrounding habitats. To be clear we précised our use of word “monitoring”. See lines x-y (a new part of the sentence underlined): “As non-essential metals are sequestered into the eggshell for excretion, the post-hatched eggshells, easy to collect in colonies may provide a convenient, non-invasive tool for monitoring (sensu stricto assessing avian exposure to habitat contamination) heavy metals contaminations in waterbirds”.

  • Have you considered analyzing eggshells from other species more common or more likely to be found in more places throughout the world? I think this would be interesting especially if these species do not travel far away from their habitats.

REPLY: As we mentioned before, all colonially breeding waterbirds seems to be a good bio-monitors of local contamination. High availability of their colonies to collects eggshells may serve as one of the main criterion of the species choice next to conservation status and susceptibility to short-time disturbance (colony visit to collect eggshells). We have added on sentence concerning this issue in the part of Discussion about limitations of the study. See lines x-y : “The colonially breeding species are the most recommended bio-monitors because one may relatively easily collect considerable sample size. Of course conservation status and susceptibility to short-term disturbance (colony visit to collect eggshells) should be considered before the choice of the species”.

Yes we did consider using other species in the World. However,the problem is samples collection requiring cooperation of many scientists from different countries as well as costly elemental analyses. It is possible to gather the data on other species from the literature positions although the results may depend on the methods. We would be delighted to broaden our research.

  • It is not clear what kind of information the proposed methodology provides for the support of decision making in the area, where the eggshells analysis is carried out. ANN may be used to predict the link between concentration of toxic element in the post-hatched eggshells of grey herons and the habitat composition in the potential foraging areas of this bird but how this information can be useful further (given the buffer of the 20km)? In line 475 you mention that “ANN may be an effective tool for wildlife and environment management and monitoring when used in the areas with similar types of main pollutants present in the environment”. This can be misleading as the contaminated food may be derived from a location near to an industrial location and not from the nearest wetland. I think conclusions should be re-written.

REPLY: This sentence does not imply indication of pollution level in wetlands or aquatic habitats. It simply says that ANN may be an effective tool for wildlife and environment management and monitoring when used in the areas with similar types of main pollutants present in the environment. However, we have changed this sentence as not the best matching here. See the new version in lines x-y: “Thus, ANN may serve asan effective tool for prediction of contamination level of waterbirds tissues based on relative area of particular habitat types in the vicinity of colonies, at least in the areas with similar types of main pollutants present in the environment. In the case of highly industrialized and/or polluted areas ANN should be adjusted for local conditions”.

The results of the research and artificial network simulations indicate the possibility of supporting decision-making processes occurring during the identification of the level of environmental contamination. In particular, the generated neural models can form the core of a dedicated information system supporting the identification process. In the future this would allow, among other things, to automate the methods of acquiring and processing information on environmental quality problems.

Reviewer 2 Report

Recommendations to authors

The idea to focus on a top predator (in freshwater ecosystems), such as the grey heron, to monitor environmental changes in essential elements, pesticides and various pollutants is great. It is also clear that the grey heron adopts the income breeder strategy for acquiring nutrients for egg production and so their maternal investments (including eggshells) should correspond to the contamination of the local breeding environment.

However, I have some questions that I think they should be answered:

  • The idea to use grey heron Ardea cinerea eggshells is great but what about the geographic areas where grey heron eggshells cannot be found? Is current study designed only for Polland?
  • Since grey herons usually travel between 1-10 km from the nesting site (and even up to 20km if attractive food is found at a greater distance), how safe is to use eggshells as biological indices? The contamination in this case may represent a location up to 20km away.
  • Obviously, herons focus on getting the food needed and they do not make choices, meaning they can get food from either a contaminated wetland or from a location near to industrial or highly populated areas. Therefore, it is difficult to deduce conclusions whether a near wetland is facing a pollution issue or not. As a result, these insights based on eggshells analysis seems unsafe to be used to determine the health of adjusting aquatic ecosystems. The sources of pollution due to human activities are already known in most cases, so what is the additional benefit of analyzing the heron eggshells?
  • What is your proposal for monitoring changes over a region of interest based on heron eggshells (collecting eggshells in different intervals)? What if eggshells are not available at a given time? Is the population of herons (or their nests) under monitoring (is it stable over time)? I think you should add a limitation section.
  • Have you considered analyzing eggshells from other species more common or more likely to be found in more places throughout the world? I think this would be interesting especially if these species do not travel far away from their habitats.
  • It is not clear what kind of information the proposed methodology provides for the support of decision making in the area, where the eggshells analysis is carried out. ANN may be used to predict the link between concentration of toxic element in the post-hatched eggshells of grey herons and the habitat composition in the potential foraging areas of this bird but how this information can be useful further (given the buffer of the 20km)? In line 475 you mention that “ANN may be an effective tool for wildlife and environment management and monitoring when used in the areas with similar types of main pollutants present in the environment”. This can be misleading as the contaminated food may be derived from a location near to an industrial location and not from the nearest wetland. I think conclusions should be rewritten.

Author Response

Answers to Reviewer 2

We thank the Reviewer for useful comments and remarks which helped us to improve the quality of our work. Please find below the detailed answers to the questions which were raised during a review. All the corrections are marked in yellow in the manuscript text. We hope that in the present form our work is suitable to be published in Sensors MDPI special issue: Identification of Bio- and Eco-Materials Using Advanced Computational Methods.

The idea to focus on a top predator (in freshwater ecosystems), such as the grey heron, to monitor environmental changes in essential elements, pesticides and various pollutants is great. It is also clear that the grey heron adopts the income breeder strategy for acquiring nutrients for egg production and so their maternal investments (including eggshells) should correspond to the contamination of the local breeding environment.

We thank very much to the Reviewer for a positive assessment of our idea of usage of a grey heron eggshells to monitor the environment. Most of the questions asked by the Reviewer concern limitations of the study.

Please find below the detailed answers to the questions. We have added a part titled Limitations of the studies to the manuscript where the answers are included.

However, I have some questions that I think they should be answered:

  • The idea to use grey heron Ardea cinerea eggshells is great but what about the geographic areas where grey heron eggshells cannot be found? Is current study designed only for Poland?

REPLY: We used here the grey heron as a model species given its common occurrence in Poland. However, other top predators from freshwater ecosystems that can bio-magnify and bio-accumulate toxic and essential elements can be used as well. Herons, in that the grey heron have wide geographical range breeding in all regions except from polar regions. We have added one sentence about this issue to the part of Discussion about limitations of the study. See lines 499-500: “In areas where the grey heron does not occur or is rare breeder, eggshells may be collected from other top predator waterbirds; the colonially breeding species are the most recommended because one may relatively easily collect considerable sample size.  

Although the study has been executed in Poland there is a possibility to broaden it to other geographic areas and other species being income breeders. The problem is however samples collection as well as costly elemental analyses. We would be delighted to broaden our research.

  • Since grey herons usually travel between 1-10 km from the nesting site (and even up to 20km if attractive food is found at a greater distance), how safe is to use eggshells as biological indices? The contamination in this case may represent a location up to 20km away.

REPLY: This 20 km range represents quite reasonable indication distance compared to the smaller species of birds. Passerines during the breeding season hold small territories, foraging within several hundred meters around the nest.

  • Obviously, herons focus on getting the food needed and they do not make choices, meaning they can get food from either a contaminated wetland or from a location near to industrial or highly populated areas. Therefore, it is difficult to deduce conclusions whether a near wetland is facing a pollution issue or not. As a result, these insights based on eggshells analysis seems unsafe to be used to determine the health of adjusting aquatic ecosystems. The sources of pollution due to human activities are already known in most cases, so what is the additional benefit of analyzing the heron eggshells?

REPLY: A signal from post-hatching eggshells represents contamination in all habitats explored by the herons. In the majority of cases they forage in aquatic ecosystems. In highly industrialized areas aquatic ecosystems are often contamination. Thus, relative area of various habitats can serve as a good predictor of all grey herons foraging habitats (not only aquatic) contamination and in turn contamination of their tissues and/or eggs.We have added one sentence to the part of Discussion about limitations of the study. See lines 494-496: “Considering opportunistic foraging of grey herons, contamination level in eggshells are representative not only for aquatic ecosystems but all habitats within the foraging range”.

Knowledge of pollution level from human activities is not always monitored in the correct way and even if it is not always all data about it are easily available. In many cases ecotoxicological studies served as the first warning about high contamination of particular area (e.g. history of massive DDT use).

  • What is your proposal for monitoring changes over a region of interest based on heron eggshells (collecting eggshells in different intervals)? What if eggshells are not available at a given time? Is the population of herons (or their nests) under monitoring (is it stable over time)? I think you should add a limitation section.

REPLY:A general definition of monitoring or bio-monitoring says that it is an assessment of exposure to environmental and industrial chemicals by measuring the amounts of the chemicals or their metabolites (biomarkers) in tissues or fluidsand does notinclude tracing changes in time. And we use this word in this original meaning. Thus, we did not mention about possible changes in the indicator population size. But as we mentioned before, the grey heron is not the only species serving as a good indicator of  surrounding habitats. To be clear we précised our use of word “monitoring”. See lines 490-496 (a new part of the sentence underlined): “As non-essential metals are sequestered into the eggshell for excretion, the post-hatched eggshells, easy to collect in colonies may provide a convenient, non-invasive tool for monitoring (sensu stricto assessing avian exposure to habitat contamination) heavy metals contaminations in waterbirds”.

  • Have you considered analyzing eggshells from other species more common or more likely to be found in more places throughout the world? I think this would be interesting especially if these species do not travel far away from their habitats.

REPLY: As we mentioned before, all colonially breeding waterbirds seems to be a good bio-monitors of local contamination. High availability of their colonies to collects eggshells may serve as one of the main criterion of the species choice next to conservation status and susceptibility to short-time disturbance (colony visit to collect eggshells). We have added on sentence concerning this issue in the part of Discussion about limitations of the study. See lines 500-501: “The colonially breeding species are the most recommended bio-monitors because one may relatively easily collect considerable sample size. Of course conservation status and susceptibility to short-term disturbance (colony visit to collect eggshells) should be considered before the choice of the species

Yes we did consider using other species in the World. However,the problem is samples collection requiring cooperation of many scientists from different countries as well as costly elemental analyses. It is possible to gather the data on other species from the literature positions although the results may depend on the methods. We would be delighted to broaden our research.

  • It is not clear what kind of information the proposed methodology provides for the support of decision making in the area, where the eggshells analysis is carried out. ANN may be used to predict the link between concentration of toxic element in the post-hatched eggshells of grey herons and the habitat composition in the potential foraging areas of this bird but how this information can be useful further (given the buffer of the 20km)? In line 475 you mention that “ANN may be an effective tool for wildlife and environment management and monitoring when used in the areas with similar types of main pollutants present in the environment”. This can be misleading as the contaminated food may be derived from a location near to an industrial location and not from the nearest wetland. I think conclusions should be re-written.

REPLY: This sentence does not imply indication of pollution level in wetlands or aquatic habitats. It simply says that ANN may be an effective tool for wildlife and environment management and monitoring when used in the areas with similar types of main pollutants present in the environment. However, we have changed this sentence as not the best matching here. See the new version in lines 5-9-512: “Thus, ANN may serve as an effective tool for prediction of contamination level of waterbirds tissues based on relative area of particular habitat types in the vicinity of colonies, at least in the areas with similar types of main pollutants present in the environment. In the case of highly industrialized and/or polluted areas ANN should be adjusted for local conditions”.

The results of the research and artificial network simulations indicate the possibility of supporting decision-making processes occurring during the identification of the level of environmental contamination. In particular, the generated neural models can form the core of a dedicated information system supporting the identification process. In the future this would allow, among other things, to automate the methods of acquiring and processing information on environmental quality problems.

Round 2

Reviewer 1 Report

Dear Authors,

I would like to thank you for your effort in improving the paper according to my comments and suggestions. Now, it looks much better. I will recommend it for publication.

Best regards,

R.